# Intravenous Magnesium Sulfate Reduces the Need for Antiarrhythmics during Acute-Onset Atrial Fibrillation in Emergency and Critical Care

**DOI:** 10.3390/jcm11195527

**Published:** 2022-09-21

**Authors:** Emanuele Gilardi, Fulvio Pomero, Enrico Ravera, Andrea Piccioni, Michele Cosimo Santoro, Nicola Bonadia, Annamaria Carnicelli, Luca Di Maurizio, Luca Sabia, Yaroslava Longhitano, Angela Saviano, Veronica Ojetti, Gabriele Savioli, Christian Zanza, Francesco Franceschi

**Affiliations:** 1Department of Emergency Medicine, Fondazione Policlinico Universitario “A. Gemelli” IRCCS, 00168 Rome, Italy; 2Department of Internal Medicine, Michele and Pietro Ferrero Hospital, 12060 Verduno, Italy; 3Foundation of “Ospedale Alba-Bra”, Department of Emergency Medicine, Anaesthesia and Critical Care Medicine, Michele and Pietro Ferrero Hospital, 12060 Verduno, Italy; 4Emergency Medicine and Surgery, IRCCS Fondazione Policlinico San Matteo, 27100 Pavia, Italy

**Keywords:** atrial fibrillation, magnesium sulfate, Flecainide, rhythm control, rate control

## Abstract

Several studies have suggested the potential role of Magnesium Sulfate (MgSO_4_) for the treatment of Atrial Fibrillation (AF) but, in clinical practice, the use of magnesium is not standardized although it is largely used for the treatment of supraventricular arrhythmias. Objectives. We evaluated the role of MgSO_4_ infusion in association with flecainide in cardioversion of patients presenting in ED with symptomatic AF started less than 48 h before. We retrospectively searched for all patients presented in ED from 1 January 2019 to 31 December 2019 requiring pharmacological cardioversion with flecainide 2 mg/kg. Ninety-seven patients met these criteria, 46 received the administration of intravenous MgSO_4_ 2 gr (Group A), and 51 did not (Group B). Among the 97 patients, the overall cardioversion rate was 85.6%, 91.3% in Group A and 80.4% in Group B. In 27 patients out of 97, the Flecainide was not administered because of spontaneous restoration of sinus rhythm of 9 pts (Group B) and 18 pts (Group A). We also found a statistical significance in the HR at the time of cardioversion between Group A (77.8 ± 19.1 bpm) and Group B (87 ± 21.7 bpm). No complications emerged. The association between MgSO_4_ and Flecainide has not yielded statistically significant results. However, in consideration of its high safety profile, MgSO_4_ administration may play a role in ED cardioversion of acute onset AF, reducing the need for antiarrhythmic medications and electrical cardioversion procedures, relieving symptoms reducing heart rate, and reducing the length of stay in the ED.

## 1. Introduction

The antiarrhythmic effects of magnesium have long been well known, since when in 1935 Zwillinger described for the first time the use of Magnesium (Mg) as an antiarrhythmic drug. In his study, he administered Mg to patients with paroxysmal tachycardia and ventricular extrasystoles [1] Subsequently, many studies described the antiarrhythmic effects of magnesium. Its use demonstrated a remarkable reversal of arrhythmias secondary to digoxin toxicity (at risk of developing ventricular arrhythmia), and other life-threatening conditions such as torsade de pointes, for which—as is known—magnesium sulfate is currently the therapy of choice [2,3,4,5]. It is therefore not surprising that worldwide many physicians use magnesium sulfate to treat different types of arrhythmias, but, in recent years, several studies are focusing especially on the possible relationship between Mg and Atrial Fibrillation (AF). Mg is involved as a cofactor of more than 300 enzyme reactions, so it plays a key role in regulating different biochemical systems, including muscle and nerve transmission [6]. It is crucial in regulating the activation of cardiac muscle, affecting depolarization by modulating the calcium channel activity, and resting membrane potential by influencing the inward rectifier potassium channel of the myocardiocytes. For this reason, there is general agreement in considering magnesium as a membrane stabilizer, as also shown by numerous studies reporting that hypomagnesemia is dangerous for the development of arrhythmias, mainly if associated with hypokalemia and alkalosis [7].

Regarding the specific relationship between Mg and AF, hypomagnesemia is itself a risk factor for atrial fibrillation, even in the absence of other cardiovascular diseases, as described in several more recent studies. In 2013, analyzing 3530 participants of the Framingham Offspring Study in a 20-year follow-up, Khan et al. [8] reported a moderated association between low serum Mg and the development of AF, with Markovits et al. [9] who found in 2016 a substantially superimposable result. Other studies, focused on the role of Mg for rhythm control in acute AF along with antiarrhythmic drugs, demonstrating an increased success of cardioversion but without a dose standardization among different studies [10]. On the other hand, the antiarrhythmic effect of Mg can also be dependent on its ability to reduce heart rate, thus achieving a rate control approach, as reported in two studies [11,12], mostly with digoxin as background therapy. Although these data suggest the potential role of magnesium as an antiarrhythmic drug for the treatment of AF, in particular for its membrane-stabilizing properties, in clinical practice the role of intravenous infusion of Mg by itself in the treatment of AF is not well known. Indeed, in a recent trial, in which patients were randomized to receive Mg or placebo before electrical cardioversion, Rajagopalan et al. [13] found that Mg infusion did not statistically increase the rate of successful cardioversion of AF, but they confirmed its excellent safety profile. In a previous study, Sultan et al. reported that a solution of magnesium and potassium administered before electrical cardioversion could reduce the required energy [14]. Therefore, despite the wide use of Mg in clinical practice, especially for its high safety profile, there is not a general agreement about the use of magnesium sulfate. There is a debate in particular about the dosage, dilution, and infusion rate of Mg and also if it can be used in association with other drugs. Even in our Emergency Department (ED), the use of magnesium is not standardized, although it is largely used for the treatment of supraventricular arrhythmias.

Our study aims to evaluate the role of magnesium sulfate infusion, alone or in association with I°C class antiarrhythmic drugs, in cardioversion of patients presenting in ED with symptomatic AF started less than 48 h before.

## 2. Materials and Methods

In this retrospective study, we searched from our computer data system for all patients presented in our ED from 1 January 2019 to 31 December 2019 with symptomatic AF started less than 48 h before. From data records it was extremely difficult to trace the exact moment of cardioversion, we chose 2-h as the cutoff for cardioversion, as previously reported in a meta-analysis [15], and 6-h as the cutoff for “non cardioversion”, because after this observation time the patients were moved from the emergency room to an area of observation.

Inclusion criteria: patient with a diagnosis of “atrial fibrillation”, for which the moment of onset was clearly highlighted and to whom the emergency physicians prescribed Flecainide 2 mg/kg given intravenously.

Exclusion criteria are any incomplete medical record, prescription of other antiarrhythmic drugs, history of congestive heart failure or structural myocardial alteration, meanings electrolytic alterations on arrival at the ED, hemodynamic instability, and dialysis patients.

Ninety-seven patients met these criteria, of which 46 received the administration of intravenous magnesium sulfate (MgSO_4_) 2 gr (Group A), and 51 did not (Group B).

### Statistical Analysis

STATA 11.0 (Stata Corp, LP, College Station, TX, USA) was used for statistical analysis. Continuous data were expressed as the mean ± standard deviation (SD). Categorical variables were presented as the number and the percentage. The groups were compared for baseline differences using test t or Chi-squared test depending upon the type of variable considered. A *p* value less than 0.05 was considered to be significant.

Data are shown as mean ± standard deviation or number (%), as more appropriate; * *p*-value for comparison between groups of patients.

## 3. Results

The main findings are reported in Table 1.

Among the 97 patients, the overall cardioversion rate was 85.6% (83 out of 97 pts), 91.3% in Group A (42 patients out of 46), and 80.4% in Group B (41 pts out of 51), without statistical significance (*p* = 0.12). Of these 83 cardioverted pts, 18 pts were cardioverted before Flecainide infusion in Group A (18/42, 42.9%), and 9 pts in Group B (9/41, 21.9%), with a *p* = 0.04.

We found that in 27 patients out of 97 the Flecainide was prescribed but not administered; this was because of spontaneous restoration of sinus rhythm during observation in 9 pts in Group B (9 out of 51, 17.6%) and in 18 patients in Group A who cardioverted after MgSO_4_ infusion (18 out of 46, 39.1%) with *p* = 0.018. Of 70 pts treated with Flecainide, 24 patients out of 28 cardioverted in Group A (85.7%) and 32 out of 42 cardioverted in Group B (76.1%), *p* = 0.3.

Analyzing patients in Group A separately (patients pretreated with 2 gr of MgSO_4_), 8/46 (17.4%) have cardioverted during MgSO_4_ infusion (less than 2 gr iv); 10/46 (21.7%) after the administration of MgSO_4_ at a dosage of 2 gr iv; 15/46 (32.6%) during the infusion of Flecainide 2 mg/kg; 5/46 (10.9%) within 2 h of the slow bolus of Flecainide and 4/46 (8.7%) within 6 h after the slow bolus of Flecainide 2 mg/kg preceded by the infusion of MgSO_4_.

The average value of the NT-proBNP on arrival in ED was significantly different (*p* = 0.01) between not-cardioverted patients (2573 pg/mL, SD ± 2972), patients cardioverted spontaneously (637 g/mL; SD ± 470) and patients cardioverted after Flecainide infusion (621 pg/mL; SD ± 819). No difference in means between Group A and Group B.

The average heart rate (HR) overlapped in the two groups of patients upon arrival in ED, 128.5 ± 23.8 bpm for Group A and 129.1 ± 23.2 bpm for Group B. After MgSO_4_ infusion, however, we reported a significant reduction in mean heart rate, even in patients with persistent AF rhythm. In the 18 patients who cardioverted without Flecainide infusion, after the magnesium infusion HR decreased by 36.3% (from 129.2 to 79.5 bpm); HR in patients cardioverted after Flecainide decreased by 14.5% (from 128.5 to 109.2 bpm); HR in non-cardioverted patients decreased by 6.3% (from 128.6 to 122.3 bpm). A significant result was the difference between the average HR on arrival and after the infusion of magnesium in patients who cardioverted with magnesium only without the infusion of antiarrhythmic, decreasing from 129.2 bpm to 71.9 bpm (*p* = 0.001). We also found a statistical significance in the HR at the time of cardioversion between Group A (77.8 ± 19.1 bpm) and Group B (87 ± 21.7 bpm), with a *p* = 0.026. See Figure 1.

No significant difference was found in electrolytes value at baseline and after MgSO_4_ infusion, except for a slight increase in the plasma concentration of magnesium, as expected, but without statistical or clinical significance.

No differences were found in gender (male vs female) for overall cardioversion. The only difference was found in home therapy with Beta Blockers, 10/53 male pts vs. 18/44 female pts (*p* = 0.17). However, we found no difference between patients treated and untreated with Beta-Blockers concerning overall cardioversion with or without the infusion of Flecainide or MgSO_4_, nor about heart rate on arrival or after cardioversion.

From the retrospective analysis of the medical records, no complications emerged during the infusion of MgSO_4_ or Flecainide.

## 4. Discussion

Symptomatic AF is a common reason for ED visits. Although an immediate restoration of sinus rhythm has not been shown to be of benefit concerning morbidity and mortality, symptomatic relief, in terms of reduction of heart rate, is an essential goal for patients [16]. Apart from the correction of underlying electrolyte abnormalities and the treatment of possible triggers (thyrotoxicosis, infections, etc.) both pharmacological and electrical cardioversion are effective in restoring sinus rhythm. In our study, we retrospectively evaluated whether a strategy of pretreatment with magnesium sulfate could achieve a high enough successful cardioversion rate, restricting Flecainide infusion only to patients not cardioverted during magnesium infusion.

We found that about 40% of patients in Group A (receiving 2 gr of MgSO_4_ i.v.) achieved successful cardioversion with MgSO_4_ infusion alone, while an overall 85.71% were cardioverted within two hours from the start of MgSO_4_ infusion, whether they required or not Flecainide infusion. The overall cardioversion rate within 6 h was 91.3%.

These data suggest that MgSO_4_ could act like an “antiarrhythmic drug” per se, maybe reducing the heart rate significantly. According to other studies, we noted that, after Mg infusion, the HR reduction in percentage was higher in cardioverted than in non-cardioverted patients. Furthermore, the HR reduction in percentage (−6.3 %) was lower in patients not responsive to the pharmacological treatment, suggesting that an early heart rate reduction could be predictive of early cardioversion.

On the other hand, we observed a high number of returns to sinus rhythm, although not statistically significant, with an overall success rate for a strategy of MgSO_4_ plus Flecainide comparable to electrical cardioversion. We also observed a surprisingly high rate of cardioversion with magnesium alone. Hence, we believe that our results lend support, albeit circumstantial, to the thesis that Mg administration may play a role in cardioversion to sinus rhythm. Given that Mg has a pathophysiological rationale, and is safe and inexpensive, we believe that our results may contribute to forming the basis for a subsequent, more reliable prospective study.

We believe that our data are encouraging for two reasons. First, MgSO_4_ infusion alone achieved successful cardioversion in 40% of patients, without the need for antiarrhythmic drug administration, thus sparing the risk for side effects of this class of medications. This rate of restoration of sinus rhythm is somewhat higher than those observed within three hours in the placebo arm of RCTs on Flecainide treatment, which ranged between 14 and 28% [17], as in our control group. Moreover, in patients in Group A, successful cardioversion was observed in 85.7% and 91.3% at two and six hours, respectively. In AF of less than 48h duration, Flecainide has been shown to have a two-hour success rate of 64% [18]. In a more recent clinical trial, in which the efficacy of vernakalant was assessed, and Flecainide was used in the control group, the two-hour success rate of Flecainide was 46% [19]. Earlier studies have reported a success rate for intravenous Flecainide of 59% [20], 56.4% [21], and 71.2% [22] at two hours, 80.4% at three hours [23], and up to 82% and 90% at 8 and 12 h, respectively [24]. A similarly high rate of cardioversion for Flecainide of 91% has been reported at 8 h after an oral loading dose [25].

We believe that Mg, acting as a membrane “stabilizer”, could promote spontaneous cardioversion, and could serve as an adjuvant or facilitator of the antiarrhythmic drugs, especially in patients in whom cardiac “remodeling” is less pronounced, and this conclusion seems to be reinforced noting that the value of NT-proBNP is significantly lower in patients who cardioverted early, as previously reported [26].

A final aspect to be discussed concerns the wider implications of our study. The use of Magnesium, with its wide safety profile, albeit in doses and administration times that will be carefully evaluated, could be extremely useful not only in the therapy of acute AF but also in the prevention of relapses; its administration in chronic could, for example, help in the long-term prevention of a first episode or for certain arrhythmic pathologies, above all in patients with specific risk factors. Moreover the success of Magnesium adjuvant therapy could be the first brick to solve the big overcrowding problem [27]. 

## 5. Conclusions

A primary demand of patients arriving at ER visits for AF is the relief of symptoms. In an acute setting, the infusion of MgSO_4_ would seem to be able to reduce the heart rate regardless of Flecainide treatment, without complications. Moreover, pretreated patients with MgSO_4_ seem to be more likely to restore sinus rhythm than patients who experience spontaneous cardioversion.

In conclusion, although the association between MgSO_4_ and Flecainide has not yielded statistically significant results, in consideration of its high safety profile MgSO_4_ administration may play a role in ED cardioversion of acute onset AF of less than 48 h duration, reducing the need for antiarrhythmic medications and electrical cardioversion procedures, relieving symptoms and reducing the length of stay in the ED.

Prospective studies with clearly defined patient inclusion criteria are needed to further confirm or refute this hypothesis.

## 6. Limitations

We acknowledge that our study has significant limitations. First of all, it is observational in nature and is a single-site study. We restricted our analysis only to patients who were deemed eligible for Flecainide administration, thus implicitly selecting patients without structural heart disease or a history of ischemic heart disease. However, the decision to administer Flecainide was made by the treating physician, without a prespecified set of criteria. More important, the decision to treat with MgSO_4_ was made by the evaluating physician, thus introducing a likely source of bias. As clinical records only incompletely report the clinician’s reason for a specific treatment, we were not able to control or account for possible biases.

In our analysis, of the patients admitted to ED for symptomatic AF, only 97 were included in the final analysis, further raising the concern that they represent a highly selected population. A selection bias is certainly present, since the inclusion criteria and the low number of patients, as well as the same retrospective nature of the study, have inevitably led to the inclusion of patients who are not only highly selected, but above all very heterogeneous in their characteristics (look, for example, at the difference in BNP values), difficult to standardize by age, home therapy and pathologies in anamnesis.

Moreover, recent studies on Mg administration have reached conflicting results. Atrial fibrillation may be the final common pathway of different underlying pathophysiological mechanisms and, thus, it has been speculated that patients’ selection criteria may significantly account for those conflicting results [28]. Hence, our inability to control for possible selection bias and confounders greatly reduces the reliability of our results.

Another major limitation of our study is that our analysis relied entirely on clinical records, which, particularly in the stressful environment of the emergency department of a large university hospital, may be incomplete. Particularly, concerns about the incompleteness of clinical records restrained us from collecting and analyzing data on comorbidities. Taken together, these limitations highlight that our results and consideration must be taken with caution.

## 7. Ethical Statement

This material is the authors’ own original work, which has not been previously published elsewhere. The paper is not currently being considered for publication elsewhere. The paper reflects the authors’ own research and analysis in a truthful and complete manner. The paper properly credits the meaningful contributions of co-authors and co-researchers. The results are appropriately placed in the context of prior and existing research. All sources used are properly disclosed (correct citation). Literally copying of text must be indicated as such by using quotation marks and giving proper reference. All authors have been personally and actively involved in substantial work leading to the paper and will take public responsibility for its content.

## Figures and Tables

**Figure 1 jcm-11-05527-f001:**
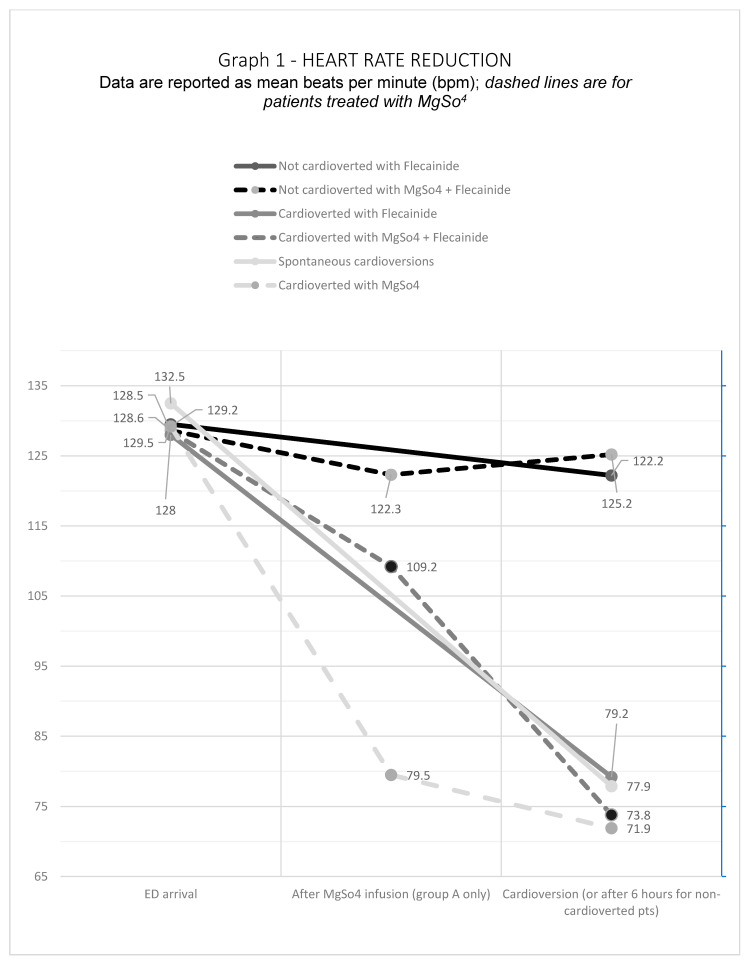
Heart Rate Reduction.

**Table 1 jcm-11-05527-t001:** Demographic Characteristics. On the LEFT, patients are classified in cardioverted vs. non-cardioverted; on the RIGHT patients are classified in MgSO_4_ infused v.sv. MgSO_4_ non-infused.

*p*-Value	14 Non-Cardioverted pts (14.4%)	83 Cardioverted pts (85.6%)	Variables	MgSO_4_ Infusion (46 pts)	No MgSO_4_ Infusion (51 pts)	*p*-Value
-	-	-	Cardioverted pts	42 (91.3%)	41 (80.4%)	*p* = 0.98
-	-	27 (32.5%)	Cardioverted before Flecainide infusion	18 (39.1%)	9 (17.6%)	*p=* 0.018
*p* = 0.43	9 (64.3%)	44 (53%)	Gender (Male)	29 (63%)	24 (47.1%)	*p* > 0.05
*p* = 0.59	64.7 (±12.1)	66.5 (±10.9)	Age	64.4 (±11.9)	67.9 (±10.1)	*p* = 0.99
*p* = 0.51	127.6 (±18.6)	121.9 (±23.9)	HR at arrival (bpm)	128.5 (±23.8)	129.1 (±23.2)	*p* = 0.99
*p* < 0.01	123.1 (±13.8) *after 6 h*	75.7 (±12.2)	HR at cardioversion (bpm)	77.8 (±19.1)	87 (±21.7)	*p=* 0.026
*p* = 0.37	2.1 (±0.2)	2.1 (±0.1)	Mg pre-Mg (mmol/L)	2.1 (±0.2)	2.1 (±0.1)	*p* = 0.87
-	-	2.8 (±0.3)	Mg post-Mg (mmol/L)	2.8 (±0.4)	-	
*p* = 0.32	4 (±0.4)	3.9 (±0.4)	K pre-Mg (mmol/L)	3.9 (±0.4)	3.9 (±0.4)	*p* = 0.93
-	-	4 (±0.5)	K post-Mg (mmol/L)	4 (±0.5)	-	
*p* < 0.01	2573.6 (±2972.4)	626.6 (±714.4)	NT-proBNP on arrival (pg/mL)	971 (±1559.7)	665 (±644)	*p* = 0.98
*p* = 0.95	5 (35.7%)	29 (34.9%)	First AF episode	20 (43.4%)	14 (27.4%)	*p* = 0.09
*p* = 0.65	5 (35.7%)	37 (44.6%)	Paroxysmal AF	19 (41.3%)	23 (45.1%)	*p* = 0.89
*p* = 0.68	1 (7.1%)	12 (14.5%)	Diuretics drugs	6 (13%)	7 (13.7%)	*p* = 0.92
*p* = 1	0	2 (2.4%)	K-sparing diuretics	2 (4.3%)	0	*p* = 0.22
*p* = 0.751	3 (21.4%)	25 (30.1%)	Beta Blockers	16 (34.7%)	12 (23.5%)	*p* = 0.22
*p* = 0.59	2 (14.3%)	8 (9.6%)	Calcium channel blockers	2 (4.4%)	8 (15.6%)	*p* = 0.09
*p* = 0.83	3 (21.4%)	20 (24.1%)	Flecainide	8 (17.4%)	15 (29.4%)	*p* = 0.16
*p* = 1	0	3 (3.6%)	Propafenone	2 (4.3%)	1 (1.9%)	*p* = 0.49
-	0	0	Digoxin	0	0	-
*p* = 1	0	1 (1.2%)	Amiodarone	1 (2.1%)	0	*p* = 0.47
*p* = 0.63	2 (14.3%)	8 (9.6%)	PPI	7 (15.2%)	3 (5.8%)	*p* = 0.13

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
