# Peer review of "Intravenous Magnesium Sulfate Reduces the Need for Antiarrhythmics during Acute-Onset Atrial Fibrillation in Emergency and Critical Care"

_jcm, 2022, doi:10.3390/jcm11195527_

Round 1

Reviewer 1 Report

This is a retrospective  non-randomized single center study focusing in highly selected patients with normal heart. The key message from my point of view is the ability of reducing AAD and or electrical CV in criticial patients using a very safe profile infusion of Mg. This is an important conclusion but it´s the rellevant message of this paper, which has to be taken with precaution since authors recognize a clear biass in patients selection.

There are some issues with need to be adressed:

Abstract: Clarify that the time to CV is measured in minutes (there is no units)

Introduction: Separate the reference number from the previous world; Consider reorganize the second paragraph, which belongs to the Discussion rather to Introduction.

Results: Consider reorganize the first paragraph, which belongs to Methods; add the unit measure (minutes) at the end of the 5th paragraph.

Author Response

Abstract: Clarify that the time to CV is measured in minutes (there is no units)

            In the abstract is not indicated the “time to CV” but the “heart rate” at the moment of cardioversion. The lack of unit is misleading.  We clarify the unit adding “beats per minute – bpm”.

Introduction: Separate the reference number from the previous world;

            The corrections indicated have been made throughout the document. Thank you.

Consider reorganize the second paragraph, which belongs to the Discussion rather to Introduction.

            This paragraph was added to “introduction” section because of its general purpose of showing what is present in the literature about the association of magnesium and atrial fibrillation. At first, it had been, as suggested, modified and added in the "discussion" section but this led to the loss of important information about the background of the study, and the "discussion" section became heavy and misleading. Leaving the paragraphs in this way seems to us to be a betterlogistical choice.

Results: Consider reorganize the first paragraph, which belongs to Methods;

            Thanks, Great suggestion for the scorrectability of the text and ease of reading.

add the unit measure (minutes) at the end of the 5th paragraph.

            As indicated, “time to cardioversion” is not indicated and numbers refer to heart rate. Units were corrected to avoid misunderstanding.

Reviewer 2 Report

            The topic of this work is important – investigating the role of magnesium sulphate infusion, alone or in association with I°C class antiarrhythmic drugs, in cardioversion of patients presenting in 89 Emergency Department with symptomatic atrial fibrillation started less than 48 hours before. This is generally a comprehensive article, and even though this study is retrospective and included a relatively small number of patients, I consider that the findings are interesting.

In order to improve the quality of the study, I have some suggestions:

1. I recommend clearly specifying the inclusion criteria and also the methodology of the study.

2. The study population is extremely inhomogeneous, and the authors should discuss this.

3. I consider that it is important to address the future scope and topics that are important and that could not be covered in the manuscript.

4. Please insert some tables and figures with the main results of the study.

Author Response

In order to improve the quality of the study, I have some suggestions:

  1. I recommend clearly specifying the inclusion criteria and also the methodology of the study.

Thanks for the indication, we tried to clarify the methodology of the study by highlighting the criteria of inclusion and exclusion.

  1. The study population is extremely inhomogeneous, and the authors should discuss this.

This is highlighted in the “limitations” section.

  1. I consider that it is important to address the future scope and topics that are important and that could not be covered in the manuscript.

Our thoughts on future implications have been made explicit at the end of the discussion. The idea is that its usefulness as a "chronic" or "maintenance" therapy for certain arrhythmic pathologies or in prevention in patients with specific risk factors can be studied prospectively and in an interventional way.

  1. Please insert some tables and figures with the main results of the study.

Tables and figures have been added to make the results clearer.
